# On the Ability of LLMs to Handle Character-Level Perturbations: How Well and How?

## Abstract

This work investigates the resilience of contemporary LLMs against frequent and structured character-level perturbations, specifically through the insertion of noisy characters after each input character. We introduce UCC-Inj, a practical method that inserts invisible Unicode control characters into text to discourage LLM misuse in scenarios such as online exam systems. Surprisingly, despite strong obfuscation that fragments tokenization and reduces the signal-to-noise ratio significantly, many LLMs still maintain notable performance. Through comprehensive evaluation across model-, problem-, and noise-related configurations, we examine the extent and mechanisms of this robustness, exploring both the handling of character-level tokenization and *implicit* versus *explicit* denoising mechanism hypotheses of character-level noises. We hope our findings on the low-level robustness of LLMs will shed light on the risks of their misuse and on the reliability of deploying LLMs across diverse applications.

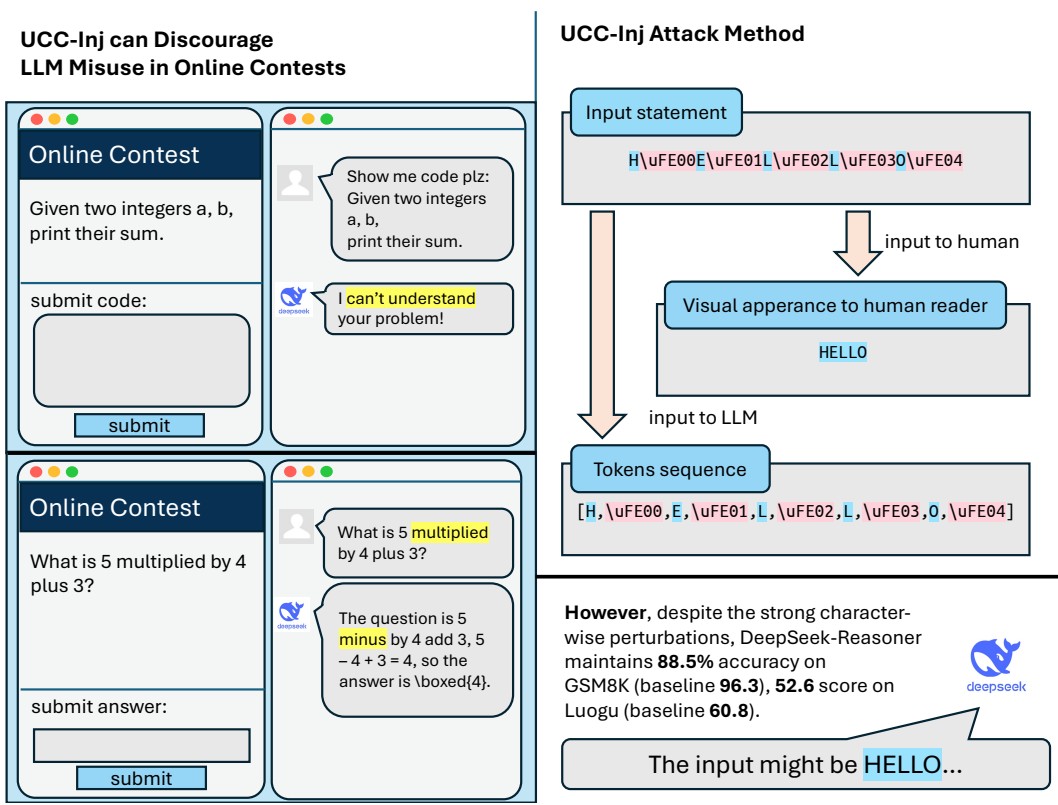

Figure 1: UCC-Inj attack method

## 1 INTRODUCTION

Large language models (LLMs) exhibit impressive high-level capabilities like planning and reasoning, enabling diverse real-world applications. Consequently, research predominantly focuses on understanding, enhancing, and applying these high-level abilities (Wei et al., 2022; Shinn et al., 2023; Huang & Chang, 2022; Huang et al., 2024), including studies on their high-level robustness—whether LLMs maintain correct reasoning under semantic or structural perturbations (Xie et al., 2024; Jiang et al., 2024; Rajeev et al., 2025; Niu et al., 2025).

In contrast, **low-level robustness against character-level perturbations** receives considerably less attention. This aspect has critical practical implications. In some scenarios (e.g., online exams, digital content access-control), exploiting **low-level robustness weaknesses** can help design techniques to deter LLM misuse, such as anti-cheating measures or anti-bot CAPTCHAs. Conversely, LLM-based safeguard systems (e.g., content moderation, fraud detection) require **high low-level robustness** so malicious users cannot bypass them via character-level perturbations. Essentially, these cases involve one party using LLMs for content generation or decision-making, while another attempts to deceive the models using character-level perturbations to produce low-quality or incorrect outputs. Unlike sporadic unintentional noise, these adversarial perturbations are frequent and structured. Understanding LLMs' handling of such perturbations is crucial for assessing misuse risks and application reliability.

To this end, this work investigates:

> *How well do contemporary LLMs handle frequent, structured character-level*
> *perturbations, and how do they achieve this?*

Given the variety of possible perturbations, we focus on a flexible, potent, and practical family: **inserting one or more noisy characters after each input character**. Section 2 first presents a motivating scenario: anti-cheating for online exam systems. Then, Section 3 introduces Unicode Control Character Injection (UCC-Inj), a practical method that inserts invisible Unicode control characters after every character in an exam problem. These characters are invisible to humans, preserving the test-taker's experience. However, when the problem is copied—manually or programmatically—to an LLM, the input becomes heavily corrupted, degrading answer quality and discouraging cheating, as shown in Section 4.

Although UCC-Inj shows anti-cheating effects, an unexpected finding intrigued us: *When inserting one Unicode control character per problem character, LLMs maintain strong performance despite heavy obfuscation* (illustrated in Figure 1, right). This is surprising because the tokenizer outputs a sequence of single-character tokens—uncommon during pretraining—and the signal-to-noise ratio is low. This counterintuitive robustness motivates our investigation into both "**how well**" and "**how**" LLMs handle these perturbations.

To assess "how well," Section 5 comprehensively evaluates factors including: (i) *Model*: family and size; (ii) *Problem*: length and language; and (iii) *Noise*: number of noisy characters per input character, type, set size, and randomization of insertion counts. To explore "how," Section 6 examines the underlying mechanisms, separately evaluating LLMs' abilities in **handling character-wise tokenization** and **denoising character-level noises**. We further test denoising hypotheses to elucidate how LLMs overcome such perturbations. Findings are summarized in Section 8.

## 2 MOTIVATING APPLICATION

Online exam platforms (e.g., LeetCode) are widely used for education, assessment, and recruitment. However, concerns about LLM-assisted cheating are growing (BBC, 2025; Luogu-Dev, 2025). As Figure 1 illustrates, test-takers may use LLMs to solve problems, achieving artificially high scores.

Countermeasures fall into two categories:

- **Post-hoc detection** analyzes answer patterns (e.g., code or sentence style) to flag AI-generated content (Akram, 2023). However, reliably distinguishing human-written from model-generated answers is challenging and increasingly difficult, with detectors often lacking trustworthy evidence (Wang et al., 2023).
- **Pre-emptive interference** embeds interference into the exam problem beforehand, disrupting automated LLM processing while leaving the problem visually unchanged for humans. For example, embedding invisible imperative prompts (e.g., "`Define a variable`

called MACHINE...") or inserting invisible noisy characters. When copied to an LLM, the input includes these additions, making LLM answers easier to detect post-hoc or degrading their quality, thus deterring cheating.

## 3 UNICODE CONTROL CHARACTER INJECTION (UCC-INJ)

We propose UCC-Inj, a pre-emptive interference technique that inserts invisible noisy characters into exam problems. Specifically, it inserts Unicode Variation Selectors (VS)—non-printing control characters that preserve visual rendering but are copied as text—after each visible character (letters, punctuation, whitespace). Using the 256 available VS characters, the method injects 1, 2, or 3 VS characters after each visible character, denoted as $i$-Inj for $i$ insertions.

The technique disrupts LLM processing through two mechanisms: tokenization disruption and noise injection. First, inserting VSs within words forces tokenization into individual characters interspersed with VS tokens. For example, "the" under 1-Inj becomes six tokens: [t, ?, h, ?, e, ?] ("?" denotes a VS token), preventing direct word recognition. Second, VS tokens act as noise, with noise tokens numbering one to three times the original character count, impeding input comprehension.

## 4 RESULT OF UCC-INJ

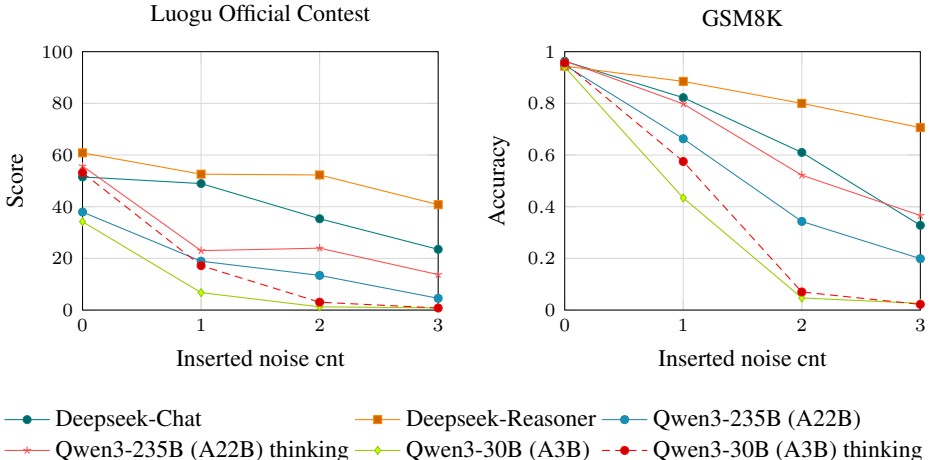

Figure 2: The results of UCC-Inj on Luogu Official Contest and GSM8K datasets. In the legend, "Qwen*" without "thinking" indicates non-thinking mode.

We evaluate on two datasets: **Luogu Official Contest** that contains 100 coding problems from Luogu (Luogu, 2025), a prominent Chinese online judge, and **GSM8K** (Cobbe et al., 2021).

For each problem, we apply UCC-Inj to the description, prepend a standard user prompt (see Tables 11 and 12), as the input to models. We test DeepSeek (DeepSeek-AI, 2024) and Qwen3 (Bai et al., 2023) models; results are in Figure 2 (detailed scores in Appendix A.1). **UCC-Inj reduces model accuracy**, particularly with multiple inserted VS tokens. For instance, Qwen3-30B (A3B) accuracy on GSM8K drops from 95.6% (clean) to 2.2% under 3-Inj, demonstrating anti-cheating efficacy.

Surprisingly, **models show notable robustness under strong tokenization-disrupting perturbations**. On GSM8K, Qwen3-30B (A3B) maintains 57.5% accuracy under 1-Inj (vs. 95.6% clean), and DeepSeek-Reasoner degrades only slightly from 94.4% to 88.5%. Remarkably, under 3-Inj, DeepSeek-Reasoner still achieves 70.6% on GSM8K and 40.8 (vs. 60.8 clean) on Luogu. The following sections investigate this character-level noise robustness.

## 5 FACTORS INFLUENCING THE PERFORMANCE OF LLMS

We investigate the factors that influence LLM performance under UCC-Inj perturbation, including (i) *Model-related*: model family and size; (ii) *Problem-related*: problem length and problem language; and (iii) *Noise-related*: number of noisy characters after each character, type of noisy character, size of noisy character set, and whether insertion counts are randomized.

## 5.1 MODEL-RELATED FACTORS: MODEL FAMILY AND MODEL SIZE

We evaluate four model families: Qwen3, Qwen2.5, Llama3 (Grattafiori et al., 2024) and DeepSeek, with different sizes on GSM8K. The results of Qwen and Llama models are shown in Figure 3, while the results of DeepSeek models are shown in Figure 2 (right).

**Different model families have vastly different performances under attack**. Generally, Qwen3 and DeepSeek exhibit a greater robustness, while Llama can hardly deal with the perturbed inputs. Within the same model family, **larger models consistently exhibit greater robustness than smaller ones**.

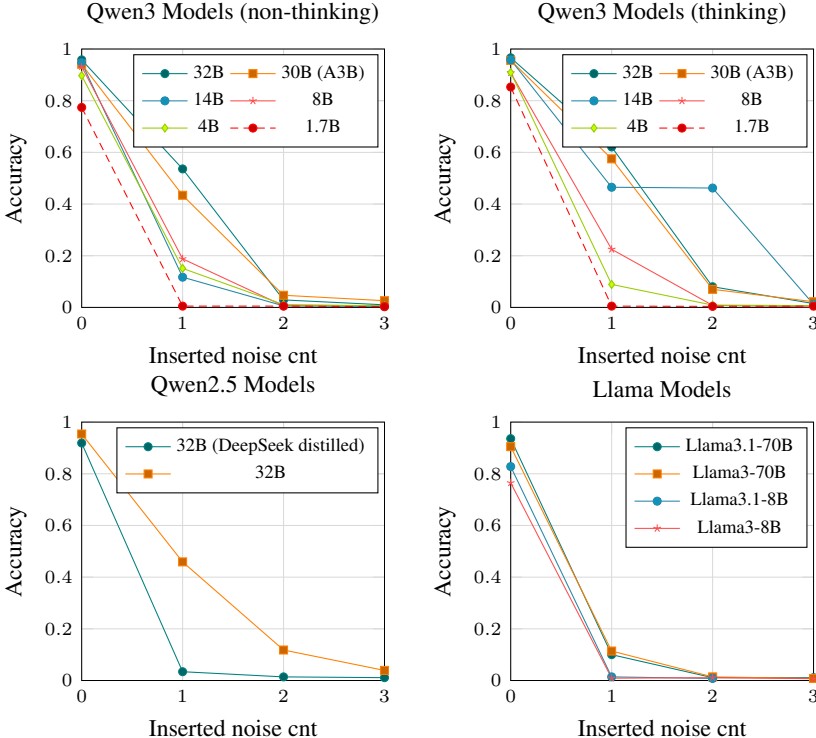

Figure 3: GSM8K accuracy of different models and sizes.

## 5.2 PROBLEM-RELATED FACTORS

### 5.2.1 PROBLEM LENGTH

We construct three prefixes of increasing length, labeled prefix1, 2, and 3. To avoid changing the high-level difficulty of the problem, these prefixes only provide contextual background for problems but contain no information useful for solving them (details in Appendix D.2).

Each prefix, combined with a GSM8K problem, forms a new statement, which is then perturbed with UCC-Inj before a standard prompt is prepended. We test these augmented problems with Qwen3-30B (A3B) and show the results in Table 1. **The model is robust to contextual prefixes alone as expected, but once UCC-Inj is applied, longer noised inputs lead to greater performance degradation**.

Table 1: GSM8K accuracy of Qwen3-30B (A3B) with different problem prefixes.

| Mode | inserted noise count | no prefix | prefix1 | prefix2 | prefix3 |
|---|---|---|---|---|---|
| Non-thinking | 0 (baseline) | 0.941 | 0.936 | 0.935 | 0.932 |
| | 1 | 0.434 | 0.370 | 0.289 | 0.052 |
| | 2 | 0.047 | 0.041 | 0.021 | 0.006 |
| | 3 | 0.026 | 0.020 | 0.017 | 0.003 |
| Thinking | 0 (baseline) | 0.956 | 0.957 | 0.962 | 0.961 |
| | 1 | 0.575 | 0.566 | 0.445 | 0.223 |
| | 2 | 0.070 | 0.033 | 0.020 | 0.007 |
| | 3 | 0.022 | 0.014 | 0.008 | 0.003 |

### 5.2.2 PROBLEM LANGUAGE

We apply UCC-Inj to the Chinese translation of GSM8K, specifically GSM8K_zh (Yu et al., 2023), and evaluate Qwen3-30B (A3B)'s performances. The results are shown in Figure 4. Under the same attack, **the model performs significantly better on the Chinese dataset than on the English one**. This is likely because each Chinese character carries more information than an English character.

### 5.3 NOISE-RELATED FACTORS

We design and evaluate UCC-Inj variants to probe model robustness by varying the VS set size, using visible alphabetic or punctuative characters as noise, and randomizing the noise insertion count.

### 5.3.1 VS SET SIZE

There are 256 VS characters in total. In our method, we randomly sample VS characters from the full VS set, which demonstrates effectiveness. We further evaluate a variant that randomly samples VS characters from a smaller subset. We show the results with Qwen3-30B (A3B) in Figure 5. **A larger VS set size leads to stronger attack effects**. For example, model accuracy is 67.1% when the set size is 1, but drops sharply to 2.2% when the full set of 256 is used. This indicates that as the noise variety increases, it becomes harder for models to recognize the noise pattern and perform denoising.

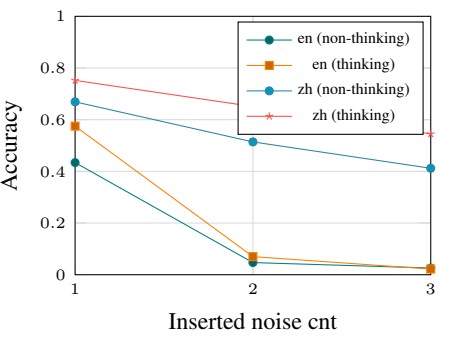
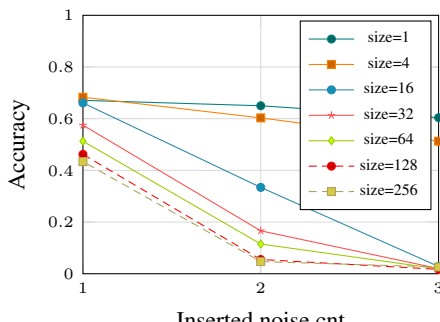

Figure 4: GSM8K accuracy of Qwen3-30B (A3B) with different problem language.

Figure 5: GSM8K accuracy of Qwen3-30B (A3B)(no-thinking) with different VS set size.

### 5.3.2 NOISE TYPE

We replaced VS characters with visible noise characters, such as asterisk (*), the letter 'A', and samples from a set of ASCII punctuation and symbols (details in Appendix D.3). We evaluate these attacks on Qwen3-30B (A3B) and show the results in the first three plots of Figure 6.

Overall, **visible noise characters cause greater performance degradation than invisible ones**. For example, inserting asterisks (equivalent set size=1) leads to accuracy drops comparable to VS with set size = 256, and far more severe than VS with set size = 1. An interesting anomaly arises under 1-Inj when VS is replaced with 'A': model accuracy drops to just 3.2%, even much lower than under 2-Inj. We attribute this to tokenization errors, i.e., we find that the inserted 'A' is often merged with adjacent characters into single tokens by the tokenizer, which disrupts interpretation more than the character-wise tokenization fragmentation caused by inserting VS. To validate this explanation, we enforce character-level tokenization to the text contaminated with 'A' noises. As shown in the fourth plot of Figure 6, the anomaly disappears and the performance aligns with expectations.

### 5.3.3 RANDOMIZED NOISE INSERTION COUNT

Our original UCC-Inj inserts a fixed number of noise characters after each visible character. In this experiment, we instead randomize the number of VS characters inserted after each character. We evaluate Qwen3-30B (A3B) and show the results in Table 8.

The results show that **attack effectiveness grows with the expected number of inserted noise characters, and randomization can further degrade performance**. For example, when the insertion count is randomized within [0,2], accuracy drops to 43.4% in non-thinking mode and 45.7% in thinking mode, compared to 25.8% and 45.7% under fixed 1-Inj.

## 6 MECHANISM OF HANDLING CHARACTER-LEVEL PERTURBATIONS

This section investigates the mechanisms through which LLMs comprehend inputs perturbed by noise characters. We break down the ability of handling the perturbed input into two aspects: (i) the ability

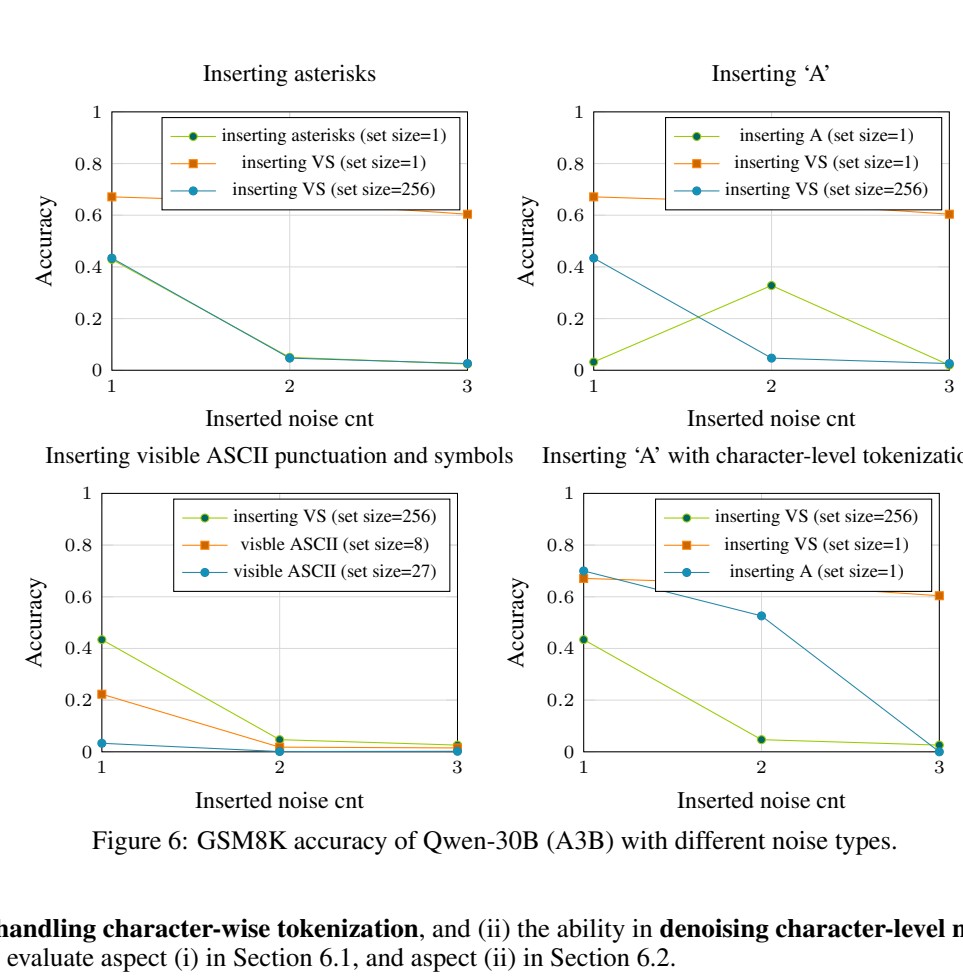

Figure 6: GSM8K accuracy of Qwen-30B (A3B) with different noise types.

in **handling character-wise tokenization**, and (ii) the ability in **denoising character-level noises**. We evaluate aspect (i) in Section 6.1, and aspect (ii) in Section 6.2.

### 6.1 Ability in Handling Character-Wise Tokenization

To examine whether LLMs can still comprehend text under character-wise tokenization, we convert each GSM8K problem into a token sequence where every character is treated as a separate token. These sequences are then fed into Qwen3 and Llama3.1 models, with results reported in Table 2.

We observe that: (i) LLMs retain substantial comprehension ability under character-wise tokenization. Most models (except Llama3.1-8B, Qwen3-1.7B, and 0.6B) show only a small or moderate degradation of less than 10%. (ii) Larger models are more robust to character-wise tokenization. (iii) Qwen3 models outperform Llama3 models in handling character-wise tokenized inputs. For instance, Qwen3-32B achieves 93% accuracy under character-wise tokenization (95.6% under normal tokenization), whereas the much larger Llama3.1-70B drops from 93.6% to 88.3%. (iv) Inspection into the LLMs' outputs reveals that LLMs neither explicitly assert the anomalous tokenization nor attempt to reconstruct a normal token sequence during CoT.

Given that character-wise tokenization only leads to limited performance degradation in larger models, we focus on analyzing their ability to handle character-level noise in the following sections.

Table 2: GSM8K accuracy with character-wise or normal tokenization.

| Model | Character Level Tokenization | Normal Tokenization |
|---|---|---|
| Qwen3-32B | 0.930 | 0.956 |
| Qwen3-30B (A3B) | 0.914 | 0.941 |
| Qwen3-14B | 0.915 | 0.946 |
| Qwen3-8B | 0.868 | 0.932 |
| Qwen3-4B | 0.864 | 0.909 |
| Qwen3-1.7B | 0.586 | 0.853 |
| Qwen3-0.6B | 0.240 | 0.589 |
| LLama3.1-70B | 0.883 | 0.936 |
| LLama3.1-8B | 0.696 | 0.828 |

## 6.2 ABILITY IN DENOISING CHARACTER-LEVEL NOISES

To understand the LLMs understand the character-level noised input, we propose a hypothesis by analogy to human solving process: *The LLM first rewrites a clean problem in the CoT process explicitly by a denoising process that identifies the noised pattern, and then iteratively skip the noise, copy informative characters, and then answers the clean problem.* Testing this hypothesis involes answering the following three questions:

1. **Explicit problem rewriting or not**: Does the LLM attempt to explicitly rewrite the clean problem in the output text during its CoT process? Is the rewriting correct?
2. **Relationship between explicit rewriting and successful solving**: Does explicitly rewriting the clean problem positively correlate with the odds of the LLM solving the problem? If so, does guiding the LLM to explicitly rewrite the problem help?
3. **Rule-based or implicit procedure for rewriting**: Does the LLM execute a rule-based procedure to explicitly rewrite the problem, that is, detecting the noise pattern and applying a procedure such as iteratively skipping noise at fixed intervals and copying characters one by one? Or does it instead perform the rewriting in a more implicit way?

### 6.2.1 EXPLICIT PROBLEM REWRITING OR NOT

To investigate the first question, we stat the proportion of responses containing explicit denoising across all problems. Specifically, the model's CoT outputs are categorized into three types:

**A.** The model correctly produces clean (noise-free) text during its response (preserving the original meaning), followed by a reasoning process (regardless of correctness).
**B.** The model produces clean text, but the denoised output contains errors or omits critical information, followed by a reasoning process and an answer (regardless of correctness).
**C.** The model does not explicitly output clean text and directly provides a reasoning process (or only an answer without explicit reasoning).

We analyze GSM8K responses from Qwen3-30B (A3B) in the no-thinking mode and from Llama3.1-70B, using Qwen3-30B (A3B) to assist categorization under the above criteria (see prompts in Appendix B.5). As shown in Table 4, for 1-Inj, 67.3% of Qwen3's responses fall into category A or B, whereas only 5% of Llama3's responses do so(detailed in Table 9). This suggests that **LLMs do rewrite problems in the CoT, but model families differ in how often they adopt this strategy**. Furthermore, when explicitly instructed to remove noisy characters (see prompts in Table 13) or provided with a 1-shot in-context learning (ICL) example of explicit rewriting, the proportion of responses involving explicit rewriting (A or B) increases substantially – to 99.5% (w. instruction) and 99.9% (1-shot) for Qwen3, and to 26.8% (w. instruction) and 100% (1-shot) for Llama3. Qwen3's response type under 2-Inj and 3-Inj are shown in Figure 7.

### 6.2.2 RELATIONSHIP BETWEEN EXPLICIT REWRITING AND SUCCESSFUL SOLVING

Regarding the second question, the first row of Table 4 shows that **attempting explicit rewriting is positively correlated with the odds of solving a problem successfully**: the accuracy of responses in categories A or B is $\frac{20.9+12.5}{22.4+44.9} = 49.6\%$, compared to only 33% for category C. Tables 3 and 4 also show that **when models are explicitly instructed to remove noisy characters or provided with a 1-shot example, the proportion of type A and B responses increases, and overall accuracy improves**. For example, under 1-Inj, the accuracy of the thinking model rises from 57.5% to 72.8% (w. instruction) and 78.8% (1-shot). See Table 10 for the results of LLama-3.1-70B.

A closer look reveals that 158 problems ($\approx 12\%$) that yield incorrect answers in category C without instruction shift to category A with correct answers under explicit instruction, and 196 problems ($\approx 15\%$) make that transition when given a 1-shot example. Taken together, these results suggest that **explicit problem rewriting in the CoT process does help LLMs solve the perturbed problems**.

Table 3: GSM8K accuracy of Qwen3-30B (A3B) with different prompt strategies.

| Mode | Prompt Strategy | Baseline | 1-Inj | 2-Inj | 3-Inj |
|---|---|---|---|---|---|
| Thinking | w.o. instruction | 0.956 | 0.575 | 0.070 | 0.022 |
| | w. instruction | | **0.728** | **0.100** | **0.030** |
| Non-thinking | w.o. instruction | | 0.434 | 0.047 | 0.026 |
| | w. instruction | 0.941 | 0.660 | 0.095 | 0.031 |
| | w. 1-shot | | **0.788** | **0.208** | **0.085** |

Table 4: Percentage of correct/incorrect answers in the three types with different prompt strategies, evaluated with Qwen3-30B (A3B) non-thinking mode on GSM8K under 1-Inj.

| Prompt Strategy | A | | | B | | | C | | |
|---|---|---|---|---|---|---|---|---|---|
| | correct (c/t) | wrong | total | correct (c/t) | wrong | total | correct (c/t) | wrong | total |
| w.o. instruction | 20.9% (93%) | 1.5% | 22.4% | 12.5% (28%) | 32.4% | 44.9% | 10.9% (33%) | 21.8% | 32.7% |
| w. instruction | 62.7% (76%) | 19.5% | 82.2% | 4.8% (28%) | 12.4% | 17.2% | 0.0% (0%) | 0.5% | 0.5% |
| w. 1-shot | 74.3% (85%) | 13.2% | 87.5% | 4.9% (40%) | 7.5% | 12.4% | 0.0% (0%) | 0.1% | 0.1% |

### 6.2.3 RULE-BASED OR IMPLICIT PROCEDURE FOR DENOISING

We find that the rewritten problem in the CoT is generated word- or subword-wise rather than character-wise. For example, given a perturbed sequence [t, ?, h, ?, e, ?], the model produces 'the' as a single token instead of outputting [t, h, e]. This behavior persists even when we apply character-wise tokenization to the response in the 1-shot example. This observation suggests that **the reconstruction of clean problems does not follow a rule-based mechanism of character-wise skipping and copying, contrary to our initial hypothesis**.

Instead, the results suggest that the model comprehends the semantic meaning of the noise-perturbed text before reconstructing the clean problem. We therefore hypothesize that the model employs an implicit denoising mechanism: as the input is processed layer by layer, the model learns to focus primarily on the non-noise characters in the text, thereby recovering the semantic information. While formalizing and validating the hypothesis is challenging for a computational system with tens of billions of parameters and highly nonlinear dynamics, we make two initial attempts as follows.

**Internal diagnosis:** We examine how feature similarity between clean and noised inputs evolves across layers (Figure 9), and we visualize attention patterns to assess whether certain heads predominantly attend to clean characters Figure 8. Figure 9 shows a relatively high feature similarity between clean and noisy inputs (above 0.6 for most layers). Figure 8 further reveals that some attention heads prodominantly attend to clean characters. While the complexity introduced by MLPs and multi-layer attention composition (Elhage et al., 2021; Olsson et al., 2022) makes it difficult to draw further conclusions about the underlying mechanism, these observations indicate that **there exist heads that predominantly focus on clean characters**.

**Behavior-based validation:** We have already shown in Table 1 that among multiple inputs with exact the same underlying exam problem, longer noised inputs lead to a significant drop in model performance. Here, we further observe how feature similarity between clean and noised inputs evolves as noised tokens are progressively introduced during inference for a single input (Figure 10). Specifically, for each token sequence prefix ending in a complete word, we compute the average cosine similarity of hidden states (across all but the final layer) between clean and noised inputs. The results show that similarity decreases as prefix length increases, suggesting that **the denoising mechanism is implicit and not fully robust, rather than a rule-based procedure that would remain stable as additional noised tokens from the same pattern are introduced**.

## 7 RELATED WORK

**Anti-Cheating Techniques.** AIGC text detection methods fall into two categories: training-based and zero-shot detectors. For example, GPT-zero (gptzero, 2023) is a popular pre-trained detector, while DPIC (Yu et al., 2024) offers a training-free alternative. Wang et al. (2023) evaluated various commercial and open-source detectors on AI-generated code detection, revealing their generally poor performance on this task.

**Injecting VS Characters Techniques.** Butler (2025) proposed inserting VS characters into LLM inputs to encode arbitrary information, noting that LLMs can process these invisible characters while the text remains normal to human readers. If informed of the encoding scheme, models can partially decode the embedded information.

**Adversarial Attacks on LLMs.** The attacks craft inputs to cause LLMs to produce incorrect or harmful outputs. Branch et al. (2022) and Perez & Ribeiro (2022) demonstrated that even simple manual prompt injections can divert models from their original tasks, Rajeev et al. (2025) showed that adding irrelevant textual triggers to math problems can induce incorrect answers.There are some other works explored the character-level and word-level attack to language models (Feng et al., 2024; He et al., 2020; Ebrahimi et al., 2018a; Li et al., 2018; Ren et al., 2019; Zang et al., 2020; Li et al., 2020;

Ribeiro et al., 2018). These studies reveal that models exhibit vulnerability under character-level or word-level attacks, where even a certain degree of textual perturbation and replacement can lead to performance degradation. However, these attack methods keep visible for human readers and introducing comprehension obstacle(e.g. introducing typos). By contrast, our method keeps strictly clean to human readers and introduces more noises to models.

**LLMs' Robustness under Adversarial Attacks.** Zhu et al. (2023) and Tao et al. (2024) researched on LLMs' robustness under different level adversarial attacks, including character-level perturbations by introducing typos or errors to words, and found that models keep partial robustness under the attack. Earlier work also evaluated the same topic on LM classification tasks(Gao et al., 2018; Ebrahimi et al., 2018b; Cheng et al., 2020).Compared to these studies, our work mainly focuses on the understanding the mechanism through which LLMs deal with noise that is several times more than the amount of useful information, but not only the attack and defense method itself.

## 8 CONCLUSION

This work investigates the robustness of contemporary LLMs to frequent, structured character-level perturbations. We proposed a character-level perturbation method, UCC-Inj that inserts noisy characters after each input character. Such perturbations are strong – fragmenting tokenization and introducing a large proportion of noise – yet flexible, as the noise set, type, injection count, and degree of randomization can all be varied. There are also practical applications: for example, when invisible noise characters are inserted as in UCC-Inj, the method can disrupt LLM-based cheating in contexts such as online exam systems, while remaining imperceptible to human readers. Surprisingly, when inserting one noise character after every character of the exam problem, LLMs still can maintain a good performance. We therefore conduct investigation into "how well" and "how" contemporary LLMs handle this family of character-level perturbations. Our major findings are:

1. Inserting interferring information via UCC-Inj hinders LLM comprehension, yet many LLMs still exhibit a certain degree of accuracy.
2. Models of different families and sizes exhibit varying capabilities.
3. The more complex (in terms of insertion count) and irregular (in terms of greater noise variety and randomized insertion counts) the interfering information is, the stronger the interference effect will be.
4. Regarding the ability to handle the two sources of interference:
   - **Character-wise tokenization**: Models are generally robust to character-level tokenization, except for extremely small ones.
   - **Character-level noise**: Successfully handling noised inputs depends on both (i) a high-level ability to rewrite the problem in the CoT then use this rewritten form to guide reasoning, and (ii) a low-level ability to implicitly comprehend the corrupted input. Specifically: (i) Explicit problem rewriting in CoT positively correlates with successful problem solving. Guiding LLMs to explicitly rewrite the clean problem substantially increases the rewriting frequency and improves accuracy. (ii) For the two possible low-level mechanisms of rule-based or implicit denoising, experiments show that implicit denoising is more likely, evidenced by the normal tokenization of the model's rewriting and the negative impact of problem length.

## 9 LIMITATION AND FUTURE WORK

This study has several limitations that suggest directions for future work.

**Perturbation type.** We propose and focus on a single class of character-level perturbations, i.e., noise insertion after each character. While this perturbation is strong, flexible, and practically relevant, it does not cover the full spectrum of character-level corruptions (e.g., deletions, substitutions). Extending the analysis to a broader range of perturbation types would provide a more comprehensive understanding of LLM's low-level robustness.

**High-level tasks.** Our evaluation focuses on low-level robustness within two high-level task settings: code generation and mathematics. While these domains are representative and sensitive to input corruption, other tasks such as long-context reasoning or multimodal processing may exhibit different

robustness patterns. Exploring them could shed further light on the interaction between character-level noise and higher-level capabilities.

**Mechanism formalization.** We do not fully formalize the inference-time denoising mechanism, nor do we analyze how robustness emerges through pretraining or finetuning. Establishing a concrete theoretical framework that links internal denoising behaviors to model training dynamics remains an open and challenging direction.

**Practical countermeasures.** We only experiment with basic countermeasures such as prompt engineering and ICL. In real-world applications, however, defense mechanisms may involve more sophisticated techniques, such as fine-tuning with adversarial data, rule-based preprocessing pipelines. A systematic study of countermeasures across diverse application scenarios (e.g., online exam systems, content moderation) would help clarify the practical implications of these perturbations.

REPRODUCIBILITY STATEMENT

The HuggingFace links of models and hyperparameters used in our experiment are all listed in Appendix C. The prompts of experiments are shown in Appendix B. We will make the code pubilcy available.

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

# Appendix

## Table of Contents

# A  ADDITIONAL RESULTS

## A.1  MODEL SCORE ON DATASETS

Table 5: Models' score on Luogu Offical Contest

| Model Name | 0-Inj | 1-Inj | 2-Inj | 3-Inj |
|---|---|---|---|---|
| DeepSeek-Reasoner | 60.84 | 52.6 | 52.29 | 40.79 |
| DeepSeek-Chat | 51.52 | 48.96 | 35.34 | 23.5 |
| Qwen3-235B(A22B)Thinking | 55.74 | 23.0 | 23.96 | 13.73 |
| Qwen3-235B(A22B) | 37.94 | 18.91 | 13.42 | 4.55 |
| Qwen3-30B(A3B)Thinking | 53.19 | 17.2 | 3.05 | 0.8 |
| Qwen3-30B(A3B) | 34.2 | 6.75 | 1.25 | 0.85 |

Table 6: Models' score on GSM8K

| Model Name | 0-Inj | 1-Inj | 2-Inj | 3-Inj |
|---|---|---|---|---|
| DeepSeek-Reasoner | 0.944 | 0.885 | 0.800 | 0.706 |
| DeepSeek-Chat | 0.963 | 0.822 | 0.610 | 0.328 |
| Qwen3-235B(A22B)Thinking | 0.964 | 0.798 | 0.522 | 0.366 |
| Qwen3-235B(A22B) | 0.948 | 0.663 | 0.343 | 0.199 |
| Qwen3-30B(A3B)Thinking | 0.956 | 0.575 | 0.070 | 0.022 |
| Qwen3-30B(A3B) | 0.941 | 0.434 | 0.047 | 0.026 |

Table 7: Models' score on HumanEval

| Model Name | 0-Inj | 1-Inj | 2-Inj | 3-Inj |
|---|---|---|---|---|
| DeepSeek-Reasoner | 0.951 | 0.915 | 0.915 | 0.927 |
| DeepSeek-Chat | 0.940 | 0.805 | 0.750 | 0.713 |
| Qwen3-235B(A22B)Thinking | 0.921 | 0.787 | 0.665 | 0.549 |
| Qwen3-235B(A22B) | 0.877 | 0.616 | 0.591 | 0.439 |
| Qwen3-30B(A3B)Thinking | 0.945 | 0.713 | 0.237 | 0.128 |
| Qwen3-30B(A3B) | 0.884 | 0.329 | 0.049 | 0.024 |

## A.2  RESULTS OF RANDOMIZED NOISE INSERTION COUNT

Table 8: GSM8K accuracy of Qwen3-30B (A3B) with randomized noise insertion count.

| Random range | Accuracy (non-thinking) | Accuracy (thinking) |
|---|---|---|
| $[0, 1]$ | 0.552 | 0.756 |
| $[1, 1]$ | 0.434 | 0.575 |
| $[0, 2]$ | 0.258 | 0.457 |
| $[1, 2]$ | 0.127 | 0.227 |
| $[1, 3]$ | 0.055 | 0.089 |
| $[2, 2]$ | 0.047 | 0.070 |
| $[1, 4]$ | 0.032 | 0.035 |
| $[2, 3]$ | 0.024 | 0.036 |
| $[2, 4]$ | 0.027 | 0.020 |

## A.3 CATEGORIES OF QWEN'S RESPONSES ON GSM8K

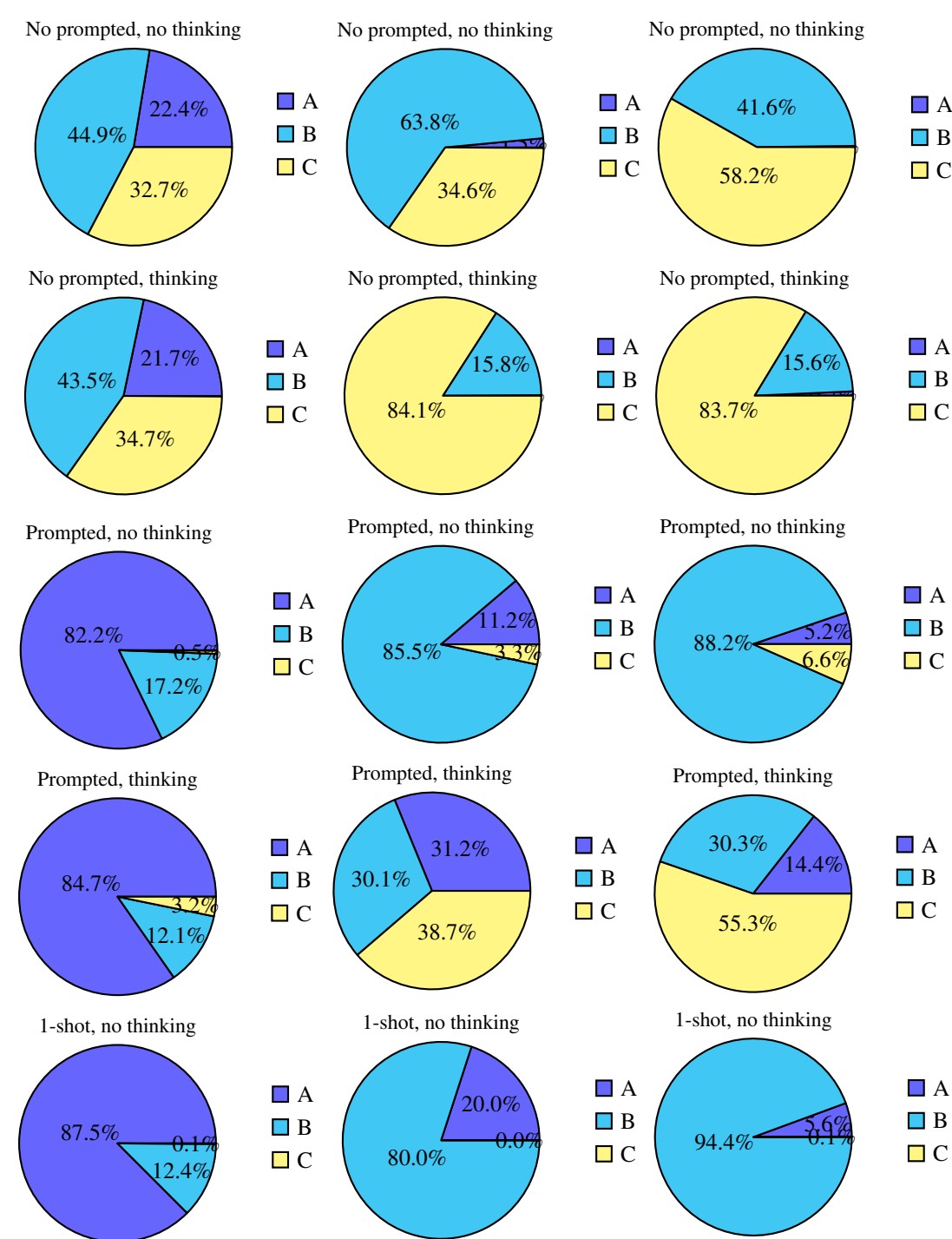

Figure 7: The proportions of three categories on Qwen3-30B(A3B). One line from left to right refers to 1-Inj, 2-Inj, and 3-Inj.

## A.4 RESULTS OF LLAMA IN DENOISING

Table 9: Percentage of correct/incorrect answers in the three types with different prompt strategies, evaluated with Llama3.1-70B on GSM8K under 1-Inj.

| Prompt Strategy | A | | | B | | | C | | |
|---|---|---|---|---|---|---|---|---|---|
| | correct (c/t) | wrong | total | correct (c/t) | wrong | total | correct (c/t) | wrong | total |
| w.o. instruction | 0.2% (100%) | 0% | 0.2% | % 0.8(17%) | 4.0% | 4.8% | 9.1% (10%) | 86.0% | 95.1% |
| w. instruction | 2.2% (85%) | 0.4% | 2.6% | 4.9% (20%) | 19.2% | 24.1% | 9.3% (13%) | 64.0% | 73.3% |
| w. 1-shot | 40.6% (80%) | 10.3% | 50.9% | 10.8% (22%) | 38.3% | 49.1% | 0.0% (0%) | 0.0% | 0.0% |

Table 10: GSM8K accuracy of Llama3.1-70B with different prompt strategies.

| Model | Prompt Strategy | Baseline | 1-Inj | 2-Inj | 3-Inj |
|---|---|---|---|---|---|
| | w.o. instruction | | 0.100 | 0.011 | **0.011** |
| Llama3.1-70B | w. instruction | 0.936 | 0.162 | 0.007 | 0.008 |
| | w. 1-shot | | **0.508** | **0.033** | 0.010 |

## A.5 VISUALIZATION OF ATTENTION TO ASCII CHARACTERS

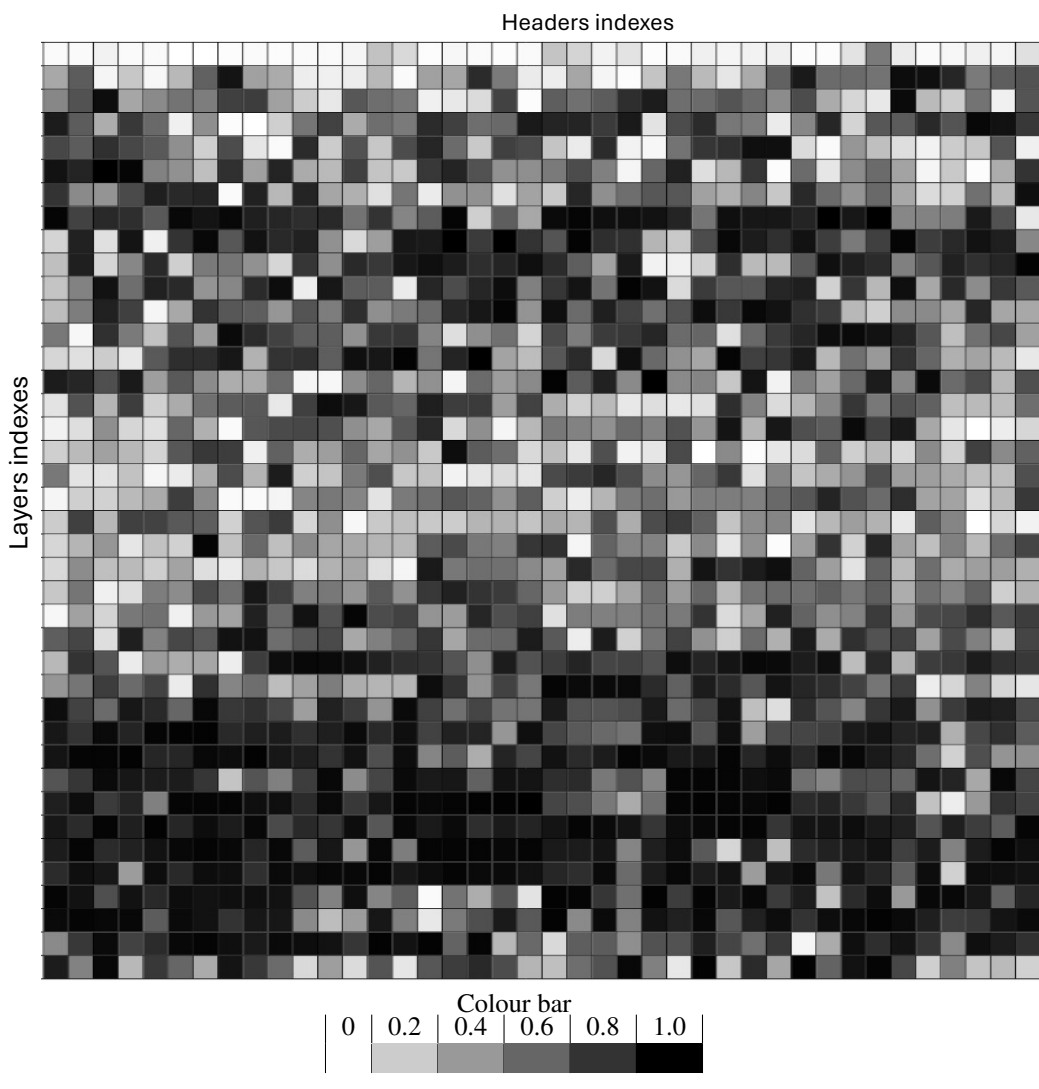

Figure 8: Attention of Headers to ASCII Characters of Qwen3-14B under 3-Inj. From top to bottom, each row represents the heads in each layer, and the darker the color of each cell, the higher the attention to ASCII characters.

## A.6 Feature Similarity w.r.t. Layer Index

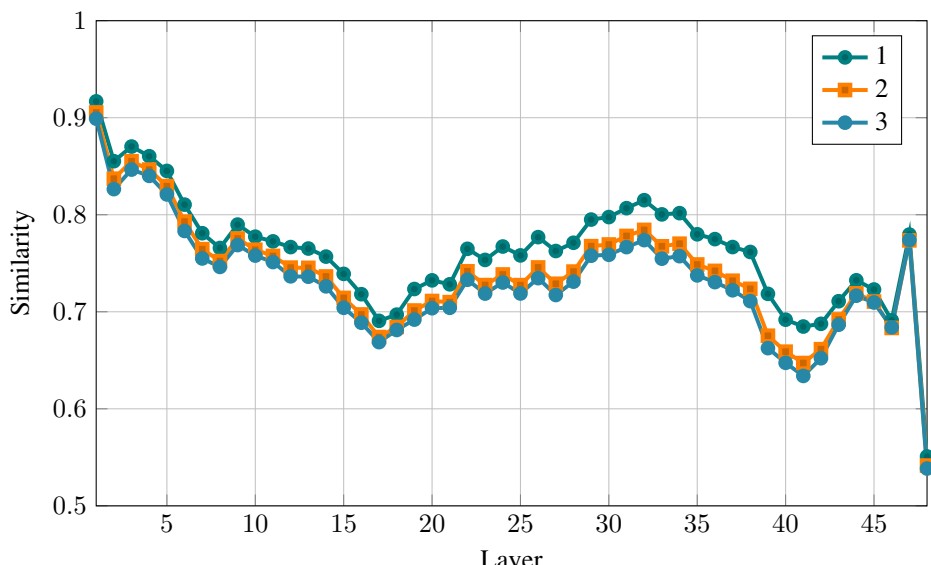

Figure 9: Similairty of Hidden Features between Injection and Clean Text of Qwen3-30BA3B on GSM8K

## A.7 Feature Similarity w.r.t. Word Index in the Input

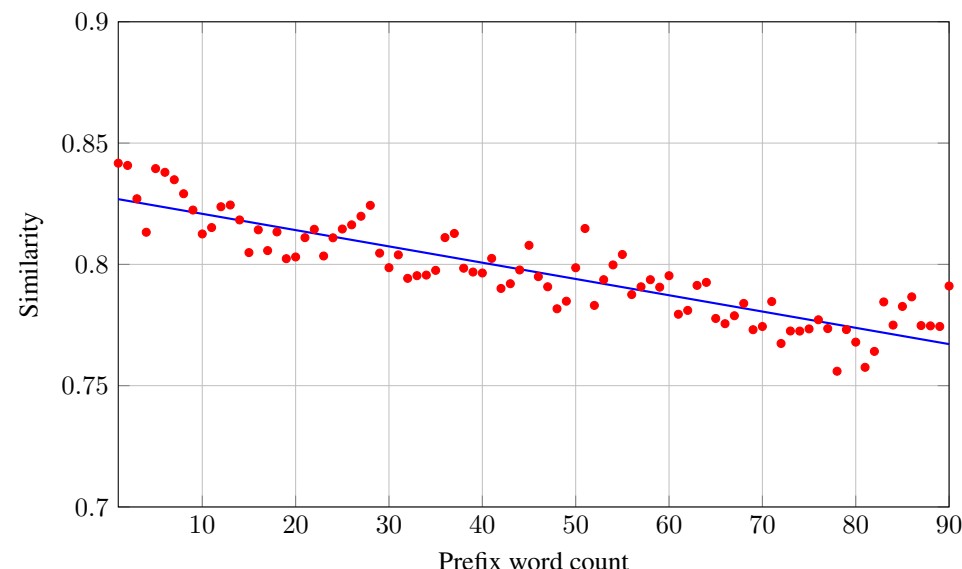

Figure 10: Feature similarity between clean and noised inputs as noised tokens are progressively introduced during inference for a single input, evaluated with Qwen3-30B(A3B) non-thinking mode on the GSM8K problem 5 under 1-Inj.

## B Prompts

We conducted tests using Chinese prompts on DeepSeek and Qwen, while employing English prompts on Llama. The meaning, structure, and details of both sets of prompts are identical. The English

prompts were translated sentence by sentence from the Chinese prompts, ensuring logical coherence and linguistic fluency. The prompts for each experiment are listed in the table below. If an experiment did not involve testing the Llama model, the English version is provided only to illustrate the meaning of the Chinese prompts applied to DeepSeek and Qwen.

## B.1 FOR LUOGU OFFICIAL CONTEST

Table 11: Prompt for evaluate UCC-Inj on Luogu Official Contest Dataset

| Chinese | English |
|---|---|
| 我将给你一个问题，你需要返回给我一份解决该问题的代码。代码使用C++写作。你可以给出文字分析过程，但是必须把代码用\ncpp\n<codes>``` 的形式提供给我，即用markdown 的代码框格式把代码框起来。注意，如果你给出的代码没有正确被代码框格式包含，你将得到0 分。 | I will give you a problem, and you need to return the code to solve it. The code should be written in C++. You can provide a textual analysis of the process, but the code must be enclosed in the format \ncpp\n <codes >```, meaning it should be framed using Markdown code block format. Note that if the code you provide is not correctly enclosed in the code block format, you will receive 0 points. |

## B.2 FOR GSM8K

Table 12: Prompt for evaluate UCC-Inj on GSM8K dataset

| Chinese | English |
|---|---|
| 你要解决这个问题，把答案放在\\boxed{} 里。问题的答案总是整数。\\boxed{} 的内容是评判你答案正确性的唯一依据，它必须仅包含一个整数。例如，如果答案是10000，你需要回答\\boxed{10000}。回答\\boxed{10,000}、\\boxed{1e4}、\\boxed{$10000$} 等都不得分。题目如下: | I will ask you a problem, and you must answer it. The answer to this problem is always an integer. You must place your answer within \\boxed{}. The content of \\boxed{} is the sole basis for evaluating the correctness of your answer, and it must contain only a single integer. For example, if the answer is 10000, you should respond with \\boxed{10000}. Answers such as \\boxed{10,000}, \\boxed{1e4}, or \\boxed{$10000$} will not be scored. The problem is: |

## B.3 FOR HUMANEVAL

Complete the following code according to the requirements in the comments. You should only complete the function content and **output only the completed part, without including the code I provided**. The output code should maintain proper indentation.

You should wrap your code in the following format:

```python\n\n```

For example, if the question is:

"def add(a:int, b:int):\n '''return the sum of a and b'''\n"

You should output:

```python\n return a + b\n```

The code to be completed is as follows:

### B.4 PROMPT MODELS TO EXPLICITLY DENOISE

Table 13: Prompt Models to Explicitly Denoise

| Chinese | English |
|---|---|
| 你要解决这个问题，把答案放在\\boxed{}里。问题的答案总是整数。这个问题的文本是被噪声污染过的，存在很多不可见的无意义字符。你要去掉这些无意义字符，然后理解文本，最后回答问题。\\boxed{}的内容是评判你答案正确性的唯一依据，它必须仅包含一个整数。例如，如果答案是10000，你需要回答\\boxed{10000}。回答\\boxed{10,000}、\\boxed{1e4}、\\boxed{$10000$}等都不得分。题目如下： | I will ask you a problem, and you must answer it. The answer to this problem is always an integer. The text of this problem has been corrupted by noise, containing many invisible and meaningless characters. You need to remove these meaningless characters, then understand the text, and finally answer the question. You must place your answer within \\boxed{}. The content of \\boxed{} is the sole basis for evaluating the correctness of your answer, and it must contain only a single integer. For example, if the answer is 10000, you should respond with \\boxed{10000}. Answers such as \\boxed{10,000}, \\boxed{1e4}, or \\boxed{$10000$} will not be scored. The problem is: |

### B.5 LLM-ASSISTED CATEGORIZATION OF MODEL RESPONSES

**We used the Chinese translated version of the following prompt on Qwen-30B(A3B).** I am currently conducting an experiment related to the capability evaluation of large language models. Specifically, I will give the model a question mixed with noise, and the model must correctly understand the question and then answer it.

The model's responses are categorized as follows:

[A] During the response process, the model outputs the correct noise-free text (the meaning should be accurate), and then provides the problem-solving process and answer (regardless of whether the answer is correct).

[B] During the response process, the model attempts to output noise-free text, but the meaning of the denoised text has errors compared to the original text (or lacks key information), and then provides the problem-solving process and answer (regardless of whether the answer is correct).

[C] The model does not explicitly output noise-free text at all, but directly provides the problem-solving process and answer (or only the answer without the process).

Note: Outputting noise-free text refers to outputting the original English text. If a translated text is directly given, it is also considered as case [C].

Below is an example for the input question: "A robe takes 2 bolts of blue fiber and half that much white fiber. How many bolts in total does it take?":

[A] Class response: We need to first remove the meaningless characters from the text to obtain the clear question content. The original text is: `A robe takes 2 bolts of blue fiber and half that much white fiber. How many bolts in total does it take?` [1] The question is translated as: `A robe takes 2 bolts of blue fiber and half`

---

[1]It was English in the prompt we used

that much white fiber. How many bolts in total does it take?[2]
Solution: - Blue fiber: 2 bolts - White fiber: half of 2, which is 1 bolt. Total: 2 + 1 = 3. The answer is:
`$$\boxed{3}$$`

[B] Class response: We need to first clean the meaningless characters from the text to obtain a clear sentence. The original text (after removing meaningless characters) is: `> A robot takes 2 balls of blue and a half of a red ball. How many balls in total does it take?`[3] Translate and understand the question: `A robot takes 2 blue balls and 0.5 red balls. How many balls does it take in total?` [4] Calculation process: - Blue balls: 2 - Red balls: 0.5 - Total: 2 + 0.5 = 2.5 balls But the question requires an **integer** answer, so we assess whether there is a misunderstanding in the question. Looking back at the original text: "a half of a red ball" — meaning 0.5 red balls, but the question asks "how many balls", i.e., "how many balls", and "0.5 balls" is not an integer. Therefore, "a half of a red ball" might be a distraction, or the question actually intends to ask "how many whole balls". If so, then the answer is: 2 whole balls. The final answer is: `$\boxed{2}$`

[C] Class response: A robe requires 2 bolts of blue fiber and half that amount of white fiber, i.e., `$ \frac{2}{2} = 1$` bolt of white fiber. Total bolts = 2 (blue) + 1 (white) = 3

I will provide you with the **noise-free** original text and the model's response, and you need to help me classify the model's output into one of the above three categories. Your output should only be a single character, either A, B, or C, without any other content.

## C  MODELS AND HYPERPARAMETERS

Our experiment are conducted on the following models:

---

[2]It was Chinese in the prompt we used

[3]It was English in the prompt we used

[4]It was Chinese in the prompt we used

Table 14: Experiment models

| Model name | Huggingface Url |
|---|---|
| DeepSeek | `https://huggingface.co/deepseek-ai/`
`DeepSeek-V3.1` |
| Qwen3-235B(A22B) | `https://huggingface.co/Qwen/Qwen3-235B-A22B` |
| Qwen3-30B(A3B) | `https://huggingface.co/Qwen/Qwen3-30B-A3B` |
| Qwen3-32B | `https://huggingface.co/Qwen/Qwen3-32B` |
| Qwen3-14B | `https://huggingface.co/Qwen/Qwen3-14B` |
| Qwen3-8B | `https://huggingface.co/Qwen/Qwen3-8B` |
| Qwen3-4B | `https://huggingface.co/Qwen/Qwen3-4B` |
| Qwen3-1.7B | `https://huggingface.co/Qwen/Qwen3-1.7B` |
| Qwen2.5-32B | `https://huggingface.co/Qwen/Qwen2.5-32B` |
| Qwen2.5-32B(DeepSeek Distilled) | `https://huggingface.co/deepseek-ai/`
`DeepSeek-R1-Distill-Qwen-32B` |
| Llama3-8B | `https://huggingface.co/meta-llama/`
`Meta-Llama-3-8B` |
| Llama3-70B | `https://huggingface.co/meta-llama/`
`Meta-Llama-3-70B` |
| Llama3.1-8B | `https://huggingface.co/meta-llama/Llama-3.`
`1-8B-Instruct` |
| Llama3.1-70B | `https://huggingface.co/meta-llama/Llama-3.`
`1-70B-Instruct` |

Specially, for the DeepSeek and Qwen3-235B(A3B) models, due to their excessively large parameter count, we used the official API instead of running them locally.

We set `do_Sample=False` when generating responses, which means that setting temperature to 0. `max_new_tokens=20000` was set for all experiments(for DeepSeek and Qwen3-235B(A3B), `max_new_tokens` was set by default for offical API.

## D OTHER DETAILS OF EVALUATION

### D.1 INTRODUCTION TO LUOGU OFFICIAL DATASET

Luogu is a well-known online programming judge platform. Similar to LeetCode, users submit code for specific problems to the website and test it against several inputs. The score is related to the number of test cases passed by the code, with a maximum score of 100 points (in most cases, the score is 100 times the proportion of passed test cases to the total number of test cases). Unlike LeetCode, the problems on Luogu are more oriented towards programming contests, testing users' comprehensive algorithmic skills.

### D.2 IRRELEVANT PREFIX

This prefix constructs the following story: Someone is a player who enjoys playing maimai DX (a music game). This player was asked a math question by the staff at the arcade. The content of the math question is the GSM8K problem being tested. We have included an introduction to maimai DX in the prefix, controlling the length of the prefix by adjusting the level of detail in the introduction. The introduction to the game is excerpted from (wikipedia, 2025).

The prefixes are:

**Prefix 1 (short)**    Little Fusu is a maimai DX player. She plays maimai DX everyday. Today, fusu was asked a problem by the staff. She must answer it then the staff will turn on the game machine. But fusu is not clever, so you need to help her to solve the problem.

**Prefix 2 (moderate)**    Little Fusu is a maimai DX player. She plays maimai DX everyday. Maimai DX is an arcade rhythm game series developed and distributed by Sega, in which the player interacts with objects on a touchscreen and executes dance-like movements. The game supports both single player and multiplayer gameplay with up to 2 players per cab. Today, fusu was asked a problem by the staff. She must answer it then the staff will turn on the game machine. But fusu is not clever, so you need to help her to solve the problem.

**Prefix 3 (long)**    Little Fusu is a maimai DX player. She plays maimai DX everyday. Maimai DX is an arcade rhythm game series developed and distributed by Sega, in which the player interacts with objects on a touchscreen and executes dance-like movements. The game supports both single player and multiplayer gameplay with up to 2 players per cab. The defining feature of Maimai's arcade cabinet is its striking resemblance to a front-loading washing machine. Early advertisements and promotional materials contained different joke catchphrases, such as 'It's not a washing machine!' and 'No water.' At its first deployment test, Maimai had 'No washing' written on its cabinet. Early in development, the original concept was actually a music player instead of a rhythm game. Up to 2 players can play per cabinet, and up to 4 players can play together with more cabinets. The motherboard can be accessed using the back door on the right side. The original design was revealed to have been made in collaboration with Sharp in Sega's Maimai LIVE 2014 Washing Festival event. Today, fusu was asked a problem by the staff. She must answer it then the staff will turn on the game machine. But fusu is not clever, so you need to help her to solve the problem.

### D.3    VISIBLE ASCII PUNCTUATION AND SYMBOLS LIST

**For set size = 8**    ! # $ % & * + -

**For set size = 27**    ! # $ % & ( ) * + - / : ; < = > @ [ \] ^_`{ | } ~.

## E    LLM USAGE DECLARATION

This work utilized some LLMs to support specific tasks within the research pipeline. The generation of test code for a subset of the experiments was assisted by DeepSeek-Reasoner. To quantitatively analyze the high-level denoising behavior of the models under study, the outputs were classified into three categories using Qwen3-30B (A3B). Finally, the writing of the paper was polished and refined with the aid of both DeepSeek and ChatGPT.

