# OpenReview forum: "On the Ability of LLMs to Handle Character-Level Perturbations: How Well and How?"
_ICLR.cc/2026/Conference — Submitted to ICLR 2026_

### Official Review · Reviewer_C1jB · 2025-10-25

**Soundness:** 3
**Presentation:** 3
**Contribution:** 2
**Rating:** 2
**Confidence:** 4

**Summary:**

## Paper Summary

This paper investigates the robustness of large language models (LLMs) against structured and frequent character-level perturbations. The authors propose **UCC-Inj**, a method that injects invisible Unicode control characters (Variation Selectors) after each input character. This technique serves dual purposes: (1) as a potential **anti-cheating mechanism** for online exam systems, and (2) as a tool for analyzing the **low-level robustness** of LLMs. The evaluation results show that inserting such Unicode control characters significantly reduces model accuracy compared with the original, non-perturbed inputs. The paper further explores *how and why* this robustness arises, examining tokenization effects, explicit and implicit denoising behaviors, and internal model representations.

**Strengths:**

## Strengths

1. **Interesting and counterintuitive findings.**
   It is surprising that LLMs retain reasoning capabilities even when tokenization is heavily fragmented. This observation provides valuable insights into LLMs’ implicit denoising abilities and internal noise resilience mechanisms.

2. **Clear and well-organized presentation.**
   The paper is well-written, logically structured, and easy to follow. The motivation, methodology, and experimental results are clearly described, allowing readers to understand both the practical and theoretical implications of the study.

3. **Mechanistic exploration beyond surface metrics.**
   The analysis goes beyond performance degradation metrics by investigating *how* models recover from noise—specifically, whether they explicitly rewrite noised inputs or rely on implicit denoising during reasoning. This adds interpretability and depth to the findings.

**Weaknesses:**

## Weaknesses

1. **Limited novelty.**
   The contribution is incremental in both problem formulation and technical design. Character-level robustness evaluation has been extensively studied in NLP, and the proposed perturbation method (Unicode control character insertion) is relatively simple and lacks methodological innovation.

2. **Narrow scope of perturbation types.**
   The paper focuses exclusively on Unicode insertion noise, without comparing other common perturbations such as substitution, deletion, or homoglyph-based attacks. This limits the generality of the findings.

3. **Unclear threat model and limited anti-cheating evaluation.**
   The description of the attacker–defender relationship in the online exam scenario is vague. No evaluation is provided to assess whether the proposed method can be easily bypassed, for instance, by filtering out Unicode control characters through simple preprocessing.

## Detailed Comments and Suggestions

1. **Clarify and strengthen novelty.**
   The concept of character-level robustness testing is not new—it has been widely explored in traditional NLP models such as LSTMs and RNNs. The authors should emphasize what *unique aspects of LLMs* are being revealed by UCC-Inj that were not observable in smaller models. The reasoning model could be counted as one of the novel subject, but the evaluation methodology and perturbation design remain largely conventional, lacking new insights or techniques that specifically leverage the reasoning characteristics of these models.
   Furthermore, the proposed method (simple insertion of invisible characters) lacks technical depth. To enhance novelty, the authors could consider generating more *adaptive* or *context-aware* perturbations that reveal specific vulnerabilities in LLM reasoning or tokenization.

2. **Extend to other perturbation types.**
   The current experiments only cover insertion noise. The authors should evaluate substitution (e.g., homoglyphs, accented characters) and deletion noise to determine whether the observed robustness is specific to insertions or more general.
   Prior work [1, 2, 3, 7, 9] on textual adversarial attacks and robustness evaluation has categorized perturbations at **three levels**—character, token, and structure. The authors should consider extending their analysis across these dimensions to make the study more comprehensive.

3. **Strengthen anti-cheating evaluation.**
   The proposed anti-cheating application is interesting but currently not convincing. Since the injected control characters can be easily removed via preprocessing scripts, it is unclear whether the defense provides real-world robustness.
   The authors should clearly define the **threat model**—including the attacker’s (cheater’s) knowledge, goals, and capabilities, and the defender’s assumptions and constraints. A security evaluation showing whether or how an “adversarial cheater” could bypass the perturbation would make the claims more credible.

4. **Improve visualization and quantitative analysis.**
   Figures 8–10 provide qualitative evidence (e.g., attention heatmaps and feature similarity curves), but the conclusions drawn from them remain anecdotal. The authors could strengthen the argument by computing **quantitative metrics**, such as KL divergence between attention distributions, CKA similarity between layer representations, or statistical comparisons of feature similarity slopes.

5. **Expand related work.**
   The related work section omits many studies on text-level and character-level robustness, adversarial noise injection, and tokenization bias. The authors should include and discuss prior work. Here are some relevant but not exhaustive related works that are missing from the paper:

[1] LLMEffiChecker: Understanding and Testing Efficiency Degradation of Large Language Models

[2] Seq2sick: Evaluating the robustness of sequence-to-sequence models with adversarial examples

[3] Structure-invariant testing for machine translation

[4] On adversarial examples for character-level neural machine translation

[5] Textbugger: Generating adversarial text against real-world applications

[6] Generating natural language adversarial examples through probability weighted word saliency

[7] Word-level textual adversarial attacking as combinatorial optimization

[8] Bert-attack: Adversarial attack against bert using bert

[9] A survey of adversarial defenses and robustness in nlp

[10] Hotflip: White-box adversarial examples for text classification

[11] Semantically equivalent adversarial rules for debugging NLP models

**Questions:**

1. **What is the threat model of the cheater?**
   The paper mentions online exam scenarios but does not clearly define the attacker’s (cheater’s) goals, knowledge, or capabilities. What assumptions are made about the cheater’s behavior and access to the system?

2. **What is the uniqueness of LLM robustness compared to traditional NLP model robustness (e.g., LSTM-based models)?**
   Since character-level robustness has been widely studied in prior NLP models, what new insights or phenomena specific to large-scale reasoning models are revealed in this work?

3. **What is the Luogu Official Contest dataset?**
   How is the evaluation metric for this dataset computed, and why did the authors choose it over more commonly used benchmarks such as HumanEval or MBPP?

4. **How practical is the proposed UCC-Inj anti-cheating defense?**
   Given that the injected Unicode control characters can be easily removed through simple preprocessing, how effective would this method remain in realistic adversarial scenarios?

5. **How can the proposed method contribute to improving model robustness?**
   Beyond serving as an evaluation or defense technique, can UCC-Inj inspire training-time defenses or model-level improvements that enhance LLM robustness against low-level perturbations?

---

> ### Author Response · Authors · 2025-11-27
>
> Thanks for your suggestions. Here is our clarification of your questions.
> 1. Threat model of the cheater. As stated in the [global response](https://openreview.net/forum?id=i3UWyQbYrq&noteId=NzbwT9k7qy), we assume cheaters are ordinary users (e.g., students) who directly copy-paste text without preprocessing, lacking technical skills to remove invisible characters.
> 2. Uniqueness of LLM robustness. Unlike traditional NLP models, LLMs exhibit implicit denoising via attention and semantic comprehension, without rule-based methods. This reveals new insights into large-scale reasoning models under extreme noise.
> 3. Luogu Official Contest dataset. Luogu is a Chinese online programming judge. Evaluation is based on code submission scores (see Appendix). We chose it for realism in exam scenarios, complementing common benchmarks.
> 4. Practicality of UCC-Inj defense. Addressed in the global response: effective against casual cheating, as preprocessing requires technical effort beyond typical users.
> 5. Contribution to model robustness. Discussed in the global response: highlights inherent robustness, inspiring training-time defenses like adversarial data augmentation for low-level perturbation resistance.
> We clarify that our method mainly focuses on the mechanism of dealing with perturbations, but not the attack method itself. You can refer our [global response](https://openreview.net/forum?id=i3UWyQbYrq&noteId=NzbwT9k7qy) for more details.

---

> > ### Comment · Reviewer_C1jB · 2025-11-28
> > **Response to authors Rebuttal**
> >
> > Thanks for your reply, after reading your response, I still have the following concerns.
> >
> >
> > [1] I am still confused about the threat model presented in the paper. The cheater appears to be treated as the victim in the current setup, yet no defensive mechanism on the cheater’s side is considered. I am concerned that this threat model may be incomplete, as the victim could simply ask an LLM to generate a filter program to remove abnormal characters.
> >
> > [2] I still do not fully understand the claimed uniqueness of LLM robustness. The perturbation noise introduced in your setting is much stronger, which does not seem to offer any advantage from the attacker’s perspective. Moreover, existing adversarial attacks have also considered models such as BERT, which similarly rely on attention layers, so the robustness challenge here does not appear fundamentally different from prior work.
> >
> > [3] After the authors’ explanation, I now understand the design of the Luogu Official Contest dataset. However, I would still like to see additional results on more commonly adopted benchmarks such as LiveCodeBench or LeetCode.
> >
> > [4] Since the threat model assumes that the victim does not employ any defensive mechanism, I continue to have concerns regarding the practicality of the proposed attack.
> >
> > [5] The authors state that “improving the robustness of LLMs is out of scope.” In that case, the main contribution of this work seems limited to demonstrating accuracy degradation on UCC-injected inputs, which may narrow its overall impact.

---

> > > ### Author Response · Authors · 2025-11-28
> > > **Response to C1jB's response-1**
> > >
> > > Many thanks for your swift response! Regarding your remaining concerns, we are happy to continue the discussion.
> > >
> > > First, we would like to emphasize: While we begin the paper with a practical motivating application, this serves only as context. Our contribution does not lie in proposing a new application scenario or a novel attack. Instead, our work focuses on understanding the phenomenon that models—somewhat unexpectedly—can handle highly noisy contexts with severe tokenization fragmentation and extremely low signal-to-noise ratios. Our goal is to analyze the mechanisms by which LLMs process such inputs, specifically examining how and how well current models cope with heavy, structured character-level perturbations.
> > >
> > > Using structured noise to “adversarially” probe the model is not only relevant to certain application scenarios, but more importantly, **serves as a tool for studying the model’s internal mechanisms**.
> > >
> > > With this clarification of the primary goals of our work, we proceed to discuss each of your questions in detail as follows:
> > >
> > > [1] **About threat Model**: We recognize that, although our work does not aim to demonstrate this application scenario nor to propose an ultra-effective attack, providing a clearer description of the scenario **through the lens of a formal threat model** would improve the clarity of the paper. We will incorporate this refinement in the revision.
> > >
> > > You are right that the cheater can be regarded as "victim" in our motivating application scenario, where the cheaters are using LLMs to answer questions in online tests. In these application scenarios, the platform wants to penalize the cheaters by obfuscating the exam problems, so that the cheaters will receive a low score if they directly rely on the generated answer of LLMs.
> > >
> > > If one wants to describe this application scenario using a threat model, the threat model goes as follows:
> > > * System setting and parties: The scenario is an online-exam/contest system. The contest organizer is the adversarial, while the participants, who are willing to cheat through LLMs are the defender.
> > > * Adversarial goal: To reduce the score that cheaters get. Meanwhile, keep the statement strictly unchanged to human readers.
> > > * Adversarial abilities: To insert invisible characters in the statement, so that the models will be perturbed.
> > > * Defender abilities: The defender will copy the statement to models, but they might find that the statement is perturbed, and ask the models to denoise it first.
> > >
> > > [2] **About the uniqueness and novelty of our work**:  In terms of perturbation design, our approach is more "strong,  flexible, and structured" compared to prior works that operate mainly at  the character level. Here, "strong" means our method introduces a  substantial amount of noise—several times greater than the effective  information in the input; "flexible" refers to the ease with which we  can control the intensity of the attack by adjusting the number of  injected characters. It is surprising that models can still process such  heavily perturbed inputs with partial comprehension, which led us to  investigate the underlying mechanism. We initially hypothesized that  models might perform rule-based denoising to recover the original  meaning. Due to the structured nature of our perturbations, we were  able to finely adjust the noise intensity and demonstrate that no such  rule-based process exists in the models. Furthermore, from a model  architecture perspective, there are significant differences between  LLMs and BERT-like models. For instance, in  BERT, we cannot explore an explicit denoising process through techniques  such as CoT rewriting.
> > >
> > > [4] **About defenses:** Indeed, in this application scenario, the cheater can incorporate countermeasures.
> > >
> > > Actually, in our paper, we have experimented with basic user countermeasures: The cheater can provide explicit instructions or even few-shot examples to guide the model in filtering input text before answering the problem. Related experiments are presented in Section 6.2.2 of the paper. Note that we did not frame these experiments from a countermeasure perspective. This is again because our focus remains on understanding intrinsic mechanism by which models remove noise, not demonstrating the effectiveness of the attack when the user incorporates defenses, or how hard the cheater can defense this "attack"/"obfuscation".

---

> > > > ### Author Response · Authors · 2025-11-28
> > > > **Response to C1jB's response-2**
> > > >
> > > > [3] **About more datasets**: We appreciate the suggestion for additional experiments. We agree that evaluating on a well-known dataset would make the work more comprehensive. Nevertheless, since the main contributions of our work lie in the analysis insights, we believe the main insights will remain intact. We have already started running experiments on HumanEval and will report back the results, and any new insights that emerge, once they are ready.
> > > >
> > > > [5] **About the contribution of our work**: We find it surprising that the models can partially understand the  input text under such strong perturbations. Our work elucidates the  mechanism by which the models denoise text under this type of attack and  reveals their capability boundaries: (i) the models can effectively  process fragmented yet noise-free tokens; (ii) the denoising ability  exhibited by the models stems from a combination of high-level text  rewriting processes and low-level comprehension of noisy tokens.  Notably, at the low level, the models do not rely on a rule-based  denoising process, and the denoising capability they demonstrate remains  implicit and unstable.

---

> > > ### Author Response · Authors · 2025-12-03
> > > **Response to C1jB's response-3**
> > >
> > > Thank you for your suggestion[3], we have supplemented the experimental results on the humaneval dataset in the appendix.

---

### Official Review · Reviewer_RAfM · 2025-10-31

**Soundness:** 2
**Presentation:** 2
**Contribution:** 2
**Rating:** 4
**Confidence:** 4

**Summary:**

The authors introduce an adversarial attack, Unicode Control Character Injection, on LLMs based on injecting invisible unicode perturbations to LLM prompts. The authors show that current state of the art LLMs are susceptible to these types of character level perturbations. They motivate the attack through an anti-cheating application, suggesting that if test questions were watermarked with this character noise students won’t be able to easily present them to LLMs to answer.

**Strengths:**

- The authors provide a novel investigation of the impact of character level noise to the response quality of LLMs.
- The authors reproduce their approach on multiple models of various sizes.

**Weaknesses:**

- It is not too surprising that performing multiple character level perturbations would break an LLMs ability to respond to user queries.
- Although the authors do introduce anti-cheat measures as a motivating application, this seems like a relatively narrow application setting. Could simply stripping copied strings of unicode characters allow users to bypass this approach?

Minor Comments

- Grammar: “To understand the LLMs understand the character”

**Questions:**

I don't have any major questions.

---

> ### Author Response · Authors · 2025-11-27
> **Response to reviewer RAfM**
>
> Thank you for pointing out our grammar error and your comments. We clarify that our main purpose of this work is to investigate the mechanism of LLMs to deal with character level perturbations. We note that models have a partial ability to understand the input statement under perturbations. We give a detailed clarification about this perspect  in the [global response](https://openreview.net/forum?id=i3UWyQbYrq&noteId=NzbwT9k7qy). We kindly refer you to our response for further elaboration.

---

### Official Review · Reviewer_HouP · 2025-10-31

**Soundness:** 2
**Presentation:** 2
**Contribution:** 2
**Rating:** 4
**Confidence:** 5

**Summary:**

The authors propose an attack (UCC-Inj) in which they insert unicode variation selectors between other text characters. They show this can induce an adversarial attack without significantly harming the model performance.

**Strengths:**

The attack is visually imperceptible in a rich text environment
The authors test the attacks over a large number of models

**Weaknesses:**

It’s easy to filter out variation selectors from a text string by specifying their unicode range, which is one line of code. This leads to a non-robust attack
Attack only works for direct copying of text in tests, which may not always be the case

**Questions:**

1. Is there a way to make the attack more robust?

---

> ### Author Response · Authors · 2025-11-27
> **Response to reviewer HouP**
>
> Thank you for your question. Actually one with coding experience can certainly easily bypass the  attack. However, our attack targets ordinary users, like normal high-school and college students. We gave a detailed clarification in our [global response](https://openreview.net/forum?id=i3UWyQbYrq&noteId=NzbwT9k7qy) and clarifed that our method has a certain degree of representativeness and practicality.
>
> Meanwhile, the main focus of our work is the LLMs' mechanism on denoising under character-level preburbation, but not the attack method itself.  We kindly refer you to our response in the [global response](https://openreview.net/forum?id=i3UWyQbYrq&noteId=NzbwT9k7qy) for further elaboration.

---

### Official Review · Reviewer_6gM2 · 2025-11-01

**Soundness:** 2
**Presentation:** 2
**Contribution:** 2
**Rating:** 2
**Confidence:** 4

**Summary:**

The paper explores the low-level robustness weakness of LLM via introducing Unicode Control Character Injection (UCC-Inj), which inserts invisible Unicode control characters in prompts.
They comprehensively evaluate considering factors of (i) Model: family and size; (ii) Problem: length and language; and (iii) Noise: number of noisy characters per input character, type, set size, and randomization of insertion counts.
They further propose the mechanism and hypotheses about how LLMs handle such perturbations.

**Strengths:**

1. The paper has a detailed presentation and empirical analysis of the factors of the attack.

2. The paper later breaks down the ability of handling the perturbed input into two aspects: (i) the ability to handle character-wise tokenization, and (ii) the ability to denoising character-level noises.

**Weaknesses:**

1. As larger models and thinking with instruction and 1-shot example are largely efficient to ease the valuneerarbiliry, I doubt if the statement "such perturbations are strong and flexible" will still hold. Also, if the method is still useful for the online examination system scenario.

2. To my knowledge, evaluating LLM's robustness to character injection/perturbation attacks has been studied before. I am unclear what the major contribution of this paper building on these literatures. For example,

[1] Kaijie Zhu, Jindong Wang, Jiaheng Zhou, Zichen Wang, Hao Chen, Yidong Wang, Linyi Yang, Wei Ye, Yue Zhang, Neil Gong, and Xing Xie. 2024. PromptRobust: Towards Evaluating the Robustness of Large Language Models on Adversarial Prompts. In Proceedings of the 1st ACM Workshop on Large AI Systems and Models with Privacy and Safety Analysis (LAMPS '24). Association for Computing Machinery, New York, NY, USA, 57–68. https://doi.org/10.1145/3689217.3690621

[2] Tao, Yiyi, et al. "Robustness of large language models against adversarial attacks." 2024 4th International Conference on Artificial Intelligence, Robotics, and Communication (ICAIRC). IEEE, 2024.

Or even earlier on LM classification tasks.

[3] Gao, Ji, Jack Lanchantin, Mary Lou Soffa, and Yanjun Qi. 2018. Black-box generation of adversarial text sequences to evade deep learning classifiers. In Proceedings of the IEEE Security and Privacy Workshops (SPW), pages 50–56.

[4] Ebrahimi, Javid, Anyi Rao, Daniel Lowd, and Dejing Dou. 2018. HotFlip: White-box adversarial examples for text classification. In Proceedings of the Annual Meeting of the Association for Computational Linguistics (ACL), pages 31–36.

...

3. The majority results of the problem-related factor analysis are not surprising, and the tested factors are mostly basic ones. It limited the paper's contribution to only providing empirical evidence.

**Questions:**

1.  Sec 6.2: The LLMs not rewriting the problem without noise doesn't necessarily mean they cannot or really don't denoise. For example, one LLM can still denoise by not attributing attention to the perturbation character tokens.

---

> ### Author Response · Authors · 2025-11-27
> **Response to Reviewer 6gM2**
>
> Thank you for your thoughtful comments.
>
> Regarding your mentioned weakness (1), we would like to clarify that when we describe our attack method as "strong and flexible," we refer to the fact that inserting invisible characters introduces a significant amount of noise—several times greater than the effective information—and that the intensity of the attack can be controlled by adjusting the quantity of injected characters. These correspond to what we mean by "strong" and "flexible," respectively. For further details on the effectiveness of this approach, please refer to the final paragraph of our response in the [global response](https://openreview.net/forum?id=i3UWyQbYrq&noteId=NzbwT9k7qy).
>
> Regarding weakness (2), we clarify that our work focuses more on the mechanism of how models handle such extensive noise, addressing the question of "how well and how." Previous studies have primarily emphasized the effectiveness of the attack methods but have offered less analysis of the underlying mechanisms by which models process these perturbations. We kindly refer you to our response in the [global response](https://openreview.net/forum?id=i3UWyQbYrq&noteId=NzbwT9k7qy) for further elaboration.
>
> Regarding your question: "The LLMs not rewriting the problem without noise doesn't necessarily mean they cannot or really don't denoise. For example, one LLM can still denoise by not attributing attention to the perturbation character tokens."
> Thank you for pointing this out. We agree that whether the model rewrites the text does not necessarily indicate whether denoising occurs. In fact, this section explores whether the model employs a rule-based denoising mechanism, not merely a rewriting mechanism. Accordingly, we have revised the title of this section to: "Rule-based or Implicit Procedure for ~~Rewriting~~ Denoising."

---

> ### Comment · Reviewer_6gM2 · 2025-11-28
> **Thank you**
>
> I thank the authors for their response. I will adjust my assessment accordingly.
>
> To Area Chairs:
>
> I hope to adjust my assessment accordingly, but I found I don't have edit access for now. Can you help check it? Thank you!

---

> > ### Author Response · Authors · 2025-11-28
> > **Thank you**
> >
> > Many thanks for your swift response and positive feedback! We appreciate your valuable comments, which have  helped us improve the work.

---

### Author Response · Authors · 2025-11-27
**Global response**

We sincerely thank all the reviewers for their valuable time and effort in reviewing our paper. We are encouraged that the reviewers acknowledge the effectiveness of the attack and the extensive examination of multiple factors that influence model performance.

Reviewer 6gM2 pointed out that prior work has studied LLMs' robustness to character-level perturbations. Reviewers HouP and RAfM noted that this attack method could be easily bypassed by using code to remove invisible characters from the input. We genuinely appreciate these thoughtful concerns and suggestions, which have helped us improve our work and led to more in-depth discussions.

In response to the comments, we would like to clarify the following aspects of our work:

1. **The major focus of our work**: Our work focuses more on understanding and explaining why and how LLMs can still partially comprehend input text despite heavy noise interference, rather than proposing the attack method itself. In other words, we mainly aim to address the question of "how well and how." Our explanations include:
   - (i) The model inherently possesses a strong ability to process fragmented but noise-free tokens, making it possible to directly understand input by focusing on fragmented visible characters.
   - (ii) If the model rewrites visible characters in the Chain of Thought (CoT) to extract meaningful information, it achieves better comprehension. Even without rewriting, the model can still understand a considerable portion of the input samples.
   - (iii) The mechanism that allows the model to "piece together" fragmented visible characters from noisy text and comprehend it at the word level is implicit and not fully robust—it is not a rule-based procedure. To achieve this capability, the model allocates more attention to visible characters.

2. **Regarding existing character-level attacks**:
Compared with prior character-level perturbation methods, our approach introduces substantially stronger noise—several times larger than the effective information in the input. We intentionally adopt this extreme setting because it is unsurprising that a model can tolerate occasional misspellings, which are likely present in its training data. In contrast, it is both unexpected and intriguing that a model can still “comprehend’’ text under our perturbations, where tokenization becomes severely fragmented and the signal-to-noise ratio is extremely low.
Although our attack is primarily designed to reveal and understand this intriguing robustness phenomenon, it also has unique practical value, as introduced in our paper. Our perturbations are completely invisible to human readers, whereas prior character-level attacks introduce visible changes (e.g., typos) that degrade human comprehension.
In summary, our attack, despite its simplicity, is exceptionally strong, structured, and exposes a surprising robustness of modern models under heavy perturbation, and differs fundamentally from existing character-level attack methods in both nature and application scenarios.
We appreciate the reviewers’ recommended references. Following these suggestions, we have added a discussion of related work and clarified that our focus is on understanding the mechanisms by which models handle noise, rather than proposing a new attack technique per se. We will further extend our analysis to include models’ mechanisms for handling previous perturbation methods in future work.

Based on the reviewers' feedback, we will reorganize our paper to make our contributions and focus clearer to readers. Additionally, our attack method has a certain degree of representativeness and practicality: our attack targets ordinary users(e.g. normal high school students), who generally lack the technical skills to easily remove noise via scripts. At the same time, there are currently no effective anti-AI cheating measures in such scenarios. Some platforms opt to inject invisible prompt injections into statements to prevent cheating. We believe that removing invisible characters is more challenging than removing hidden prompts visible in the source code. Beyond the attack itself, this work also highlights, from a defensive perspective, that the model inherently possesses some ability to resist attacks, which is valuable for studying the security of large language models. For instance, if LLMs are used to detect whether users are inputting harmful content on social media, this study reveals that even without targeted defenses against invisible character insertion (and related unknown character-level attacks), the model still exhibits a degree of robustness and has the potential to identify and reject such content.

---

### Meta-Review · Area_Chair_7mG6 · 2026-01-10

**Summary:**

The paper was evaluated for rejection, with reviewer scores concentrated in the low-to-borderline range (2 and 4). Reviewers recognized the paper’s strengths, including its comprehensive experimental evaluation, clear organization, and the intriguing finding that LLMs can retain partial understanding even under extremely noisy inputs. Nonetheless, they expressed significant concerns regarding the limited novelty relative to prior robustness studies, an unclear and weakly justified threat model, and the restricted focus on a single type of perturbation. Consequently, the contribution was seen as largely empirical and incremental, without sufficient evidence that it uncovers fundamentally new robustness behaviors unique to LLMs. Reviewers suggested that the work could be strengthened by incorporating rebuttal clarifications into the main text, more clearly framing the contribution as a mechanistic study of robustness, refining or downplaying the application-oriented threat model, broadening the evaluation to additional perturbation types and standard benchmarks, and deepening the analysis to better differentiate LLM-specific phenomena from existing literature.

**Reviewer Concerns:**

The rebuttal effectively clarified the authors’ primary intent, that the work is not proposing a novel attack or defense but rather using structured, extreme noise as a probing tool to study implicit denoising and robustness mechanisms in LLMs. Several reviewers’ confusion about the meaning of “strong and flexible” perturbations was resolved by explicitly defining these terms in relation to noise magnitude and controllability. The authors also convincingly explained that rewriting or CoT behavior should not be conflated with explicit denoising, and they revised section framing accordingly. Additional clarifications regarding the Luogu dataset, the role of user instructions and few-shot prompting, and the distinction from prior work focused purely on attack effectiveness improved the technical understanding for some reviewers.

Despite these clarifications, substantive issues remain. Several reviewers continued to question the novelty of the work relative to extensive prior research on character-level robustness and adversarial perturbations, including work on attention-based models. The uniqueness of the observed robustness in LLMs, as opposed to earlier NLP models, was not fully convincing to all reviewers. The anti-cheating motivation remained weak due to an incomplete or arguably unrealistic threat model, given that simple preprocessing could remove the injected characters. Finally, the evaluation scope was considered narrow, with limited comparison to other perturbation types or widely used benchmarks, reducing the perceived generality and impact of the findings.

**Reviewer Scores:**

Reviewer 6gM2: Likely increased from 2 → 4, reflecting improved clarity about the paper’s goals and mechanisms, but still with reservations about novelty.

Reviewer HouP: Likely remained around 4, as concerns about easy bypass and limited robustness persist despite clarifications.

Reviewer RAfM: Likely remained at 4, with continued perception that the findings are unsurprising and the application narrow.

Reviewer C1jB: Likely remained at 2, as key concerns about threat model completeness, novelty, and practical relevance were not fully resolved.

---

### Decision · Program_Chairs · 2026-01-26

Reject